# EVO-RDESIGN: LEVERAGING EVOLUTIONARY PRIORS FOR STRUCTURE-BASED RNA DESIGN

## ABSTRACT

Designing RNA sequences based on RNA tertiary structures is a crucial aspect of future RNA design with significant potential to aid drug development. Recently, deep learning-based methods have made progress in this area; however, these methods are constrained by the limited availability of RNA structural data, making it challenging to achieve optimal performance. In this paper, we propose EVO-RDesign, which leverages the evolutionary priors embedded in extensive sequence data to facilitate better RNA sequence design. Specifically, RNA language models have recently been demonstrated to learn the evolutionary information of RNA. Therefore, we consider RNA language models as repositories of evolutionary priors and design a series of adaptors that enable EVO-RDesign to retrieve these priors conditioned on the input RNA structural information. To achieve better performance, the adaptor innovatively inputs RNA structural information and outputs from existing RNA design methods into the language model. Experiments demonstrate that EVO-RDesign outperforms RDesign, achieving a **3.5%** increase in sequence recovery on RNAsolo. It also exhibits zero-shot generalization, with gains of **5.1%** and **4.1%** in sequence recovery on RNA-Puzzles and Rfam, respectively. We also apply in-silico folding to validate whether the generated sequences can fold into the specified 3D RNA backbones.

## 1 INTRODUCTION

RNA is a vital biological macromolecule that plays a crucial role in various biological processes. Since functional RNA molecules are essential for regulatory processes and transcription control, many RNA-based innovations are at the forefront of biotechnology, including mRNA vaccines and CRISPR-based genomic medicine. In recent years, the design of RNA molecules has been attracting widespread interest in fields such as therapeutics, synthetic biology, and bioinformatics.

The RNA sequence consists of four types of nucleotides, which undergoes a hierarchical folding process, as shown in Fig.1. The secondary structure of RNA is formed through hydrogen-bond patterns between the nucleotides, and these interactions are then critical in driving the folding of RNA sequence into its 3D tertiary structure.

Since the intricate and diverse functions of RNA are determined by its 3D tertiary structure, one fundamental challenge of RNA design is to generate RNA sequences that can fold into the desired structure, which is known as structure-based RNA design or RNA inverse folding.

Recently, some works (Tan et al., 2023; Joshi et al., 2024) have attempted to use geometric deep learning to address tertiary structure-based RNA design, yielding some progress. Specifically, these methods utilize RNA structure-sequence pair data as the training set. This training data is used to train a Graph Neural Network (GNN) (Zhou et al., 2020) that encodes RNA 3D structures, enabling the GNN to predict RNA sequences based on their structures.

However, these methods still face significant bottlenecks: the known RNA structure data is very limited. The number of available RNA structures is less than 1% of the number of protein structures (Adamczyk et al., 2022), primarily due to RNA's inherent instability and rapid degradation in vitro (Rother et al., 2011). This results in insufficient training data, making it challenging for deep learning methods to achieve satisfactory performance.

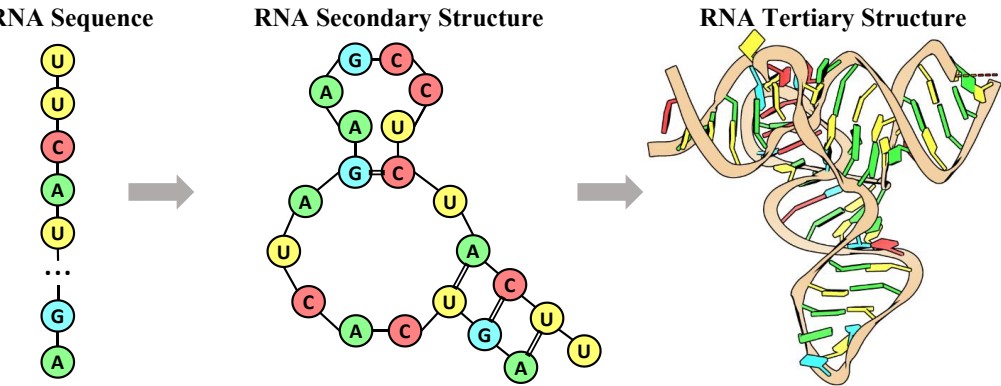

Figure 1: **RNA structures.** RNA consists of four types of nucleotide. RNA sequence can be folded into secondary structure and further tertiary structures. Single solid lines represent phosphodiester bonds, and double solid lines represent hydrogen bonds here.

To alleviate this bottleneck, we propose EVO-RDesign. The high-level insight is straightforward: while RNA structure data is limited, RNA sequence data is abundant; moreover, the evolutionary priors encoded within these sequence data may be leveraged to enhance RNA design performance. Specifically, language models trained on large-scale RNA sequence data have been shown to encode evolutionary priors (Chen et al., 2022; Wang et al., 2023). Therefore, we employ an RNA language model as an evolutionary prior knowledge base, and then use the input RNA structure to conditionally retrieve the corresponding priors to improve the design results.

To enable retrieval of corresponding priors based on RNA structure, we propose a framework and corresponding adapters that allow structural information to be mapped into a format acceptable by the language model. Furthermore, we fine-tune the language model to predict RNA sequences directly based on the input structural information. Specifically, within this framework, an input structure is first fed into an existing RNA design model, RDesign (Tan et al., 2023). Subsequently, RDesign extracts structural features of the RNA and outputs sequence design results. These structural features and design results are subsequently fed into a module called the Structural Adapter, which translates them into representations that the RNA language model can accept. These translated representations are then input into the language model to retrieve the corresponding priors. In addition to the Structural Adapter module, we map the hidden features from RDesign through a linear layer into a pairwise matrix, which serves as an attention bias added to the RNA language model. This integration allows the language model to receive RNA structural information via the attention bias.

Our contributions can be summarized as follows:

- We propose, for the first time, the idea of using evolutionary priors in RNA sequence data to assist structure-based RNA design, addressing the issue of insufficient structural data.
- We introduce the EVO-RDesign framework and a series of adaptors that enable language models incorporating evolutionary priors to design and refine RNA sequence results based on structural information.
- Extensive experiments demonstrate that EVO-RDesign significantly outperforms existing methods in terms of sequence recovery rate, surpassing RDesign by 3.5% on RNAsolo. In terms of zero-shot generalization, EVO-RDesign exceeds RDesign by 5.1% on RNA-Puzzles and 4.1% on Rfam. Additionally, EVO-RDesign exhibits strong generalization capabilities across different lengths and types of RNA.

## 2 RELATED WORK

### 2.1 STRUCTURE-BASED RNA DESIGN

Early research in computational RNA design focused solely on secondary structure constraints. Pioneering efforts like RNAinverse (Hofacker et al., 1994) and RNAiFold (Garcia-Martin et al.,

2013) utilized energy-based local search algorithms for compatibility with these secondary structures. MCTS-RNA (Yang et al., 2017) employed Monte Carlo tree search to explore viable designs. Additionally, aRNAque (Merleau & Smerlak, 2022) and eM2dRNAs (Rubio-Largo et al., 2023) introduced evolutionary algorithms specifically tailored for secondary structure-based RNA design. Recently, the field of RNA design has greatly benefited from the emergence of data-driven paradigms and geometric deep learning. LEARNA (Runge et al., 2018) attempted to tackle the challenge of secondary structure-based design using reinforcement learning. However, without considering 3D structures of RNA, previous methods cannot meet accurate functional structure constraints, thus restricting their practical application. In response, RDesign (Tan et al., 2023) and gRNAde (Joshi et al., 2024) have been developed as geometric RNA design pipelines that specifically address the challenges of tertiary structure-based RNA design.

## 2.2 Structure-based Protein Design

Similar to RNA, structure-based protein design, also known as protein inverse folding, aims to generate an amino acid sequence that will fold into a given protein backbone structure (Yue & Dill, 1992). Since the higher-order structures of RNA and proteins can be similarly described (Rother et al., 2011), the field of protein design has exhibited a progression similar to that of RNA design. Originally, structure-based protein design was tackled by physics-based energy minimization (Street & Mayo, 1999; Alford et al., 2017). Recently, it has benefited from the advantages of deep learning (Ingraham et al., 2019a; Hsu et al., 2022), including the application of protein language model (Verkuil et al., 2022; Mao et al., 2023; Zheng et al., 2023).

## 2.3 RNA Language Model

Self-supervised pre-trained language models have demonstrated their ability to extract knowledge from vast amounts of unannotated data, particularly in the realm of NLP (Devlin et al., 2019; Floridi & Chiriatti, 2020). This progress has paved the way for the application of language models to the modeling of biomacromolecules, such as DNA, RNA, and proteins, given their inherent linear sequence-based primary structure. The biological community's proactive efforts in standardizing high-quality biomolecular data (Watson, 1990; Suzek et al., 2007) contributes to the advancements of language model in the fields of DNA (Nguyen et al., 2023; Schiff et al., 2024) and proteins (Lin et al., 2022). Likewise, with the development of high-throughput RNA sequencing technologies, the RNA field has witnessed the emergence of language models designed for specific types and functions (Yang et al., 2023; Chu et al., 2024; Chen et al., 2023; Akiyama & Sakakibara, 2022), as well as general-purpose, large-scale language models (Chen et al., 2022; Wang et al., 2023). These models are now capable of performing sequence-to-structure downstream tasks, suggesting that structural information is embedded within sequence data. However, the potential of utilizing RNA language models for structure-to-sequence tasks remains an unexplored avenue.

## 3 Method

As shown in Fig.2, we design EVO-RDesign, a model-agnostic and generic approach that repurposes RNA language model for structure-based RNA design. In the following sections, we first outline the preliminaries (Sec.3.1), including a formal description of the structure-based RNA design problem and the RNA language model. Subsequently, we introduce the structure module (Sec.3.2), detailing the RNA tertiary structure modeling and initial structure-to-sequence prediction. Finally, we present a concise description of the methods we propose for sequence refinement (Sec.3.3).

## 3.1 Preliminaries

**Problem Statement.** Structure-based RNA design, also known as RNA inverse folding, involves generating functional RNA sequences that are capable of folding into a specified desired structure. Formally, the RNA inverse folding problem is defined by the mapping:

$$\mathcal{F}_\theta : \mathcal{X}^N \mapsto \mathcal{S}^N \tag{1}$$

where $\mathcal{F}_\theta$ represents a learnable function parameterized by $\theta$. For an RNA molecule consisting of $N$ nucleotides, $\mathcal{S}^N = \{s_i \in \{A, U, C, G\} \mid 1 \leq i \leq N\}$ specifies the sequence composed

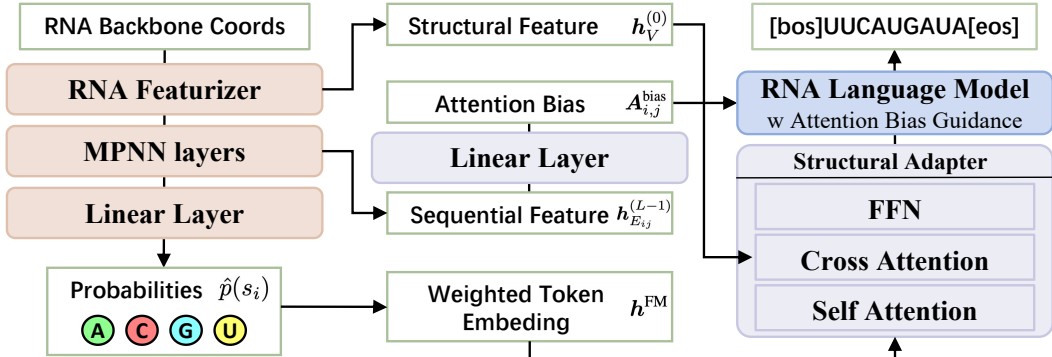

Figure 2: **Overview of EVO-RDesign.** The orange background indicates the frozen RDesign model, the blue background indicates the frozen RNA language model, and the purple background indicates trainable module.

of the four types of nucleotides. Meanwhile, the set $\mathcal{X}^N = \{\boldsymbol{x}_i^\omega \in \mathbb{R}^3 \mid 1 \leq i \leq N, \omega \in \{P, O5', C5', C4', C3', O3'\}\}$ represents the 3D coordinates of the six backbone atoms for each nucleotide. The goal of the learning process is to identify the model parameter $\theta$ that maximizes the conditional log-likelihood $p(\mathcal{S}^N \mid \mathcal{X}^N; \theta)$ based on paired RNA structure-sequence data.

**RNA Language Model.** Language models generally approximate the sequence distribution $p(\mathcal{S}^N)$ by maximizing $\prod_i p(S_i \mid S_{-i})$ over partially corrupted sequences (Devlin et al., 2019; Zheng et al., 2023). We choose RNA-FM (Chen et al., 2022), a general-purpose, large-scale, self-supervised pre-trained RNA language model, to facilitate structure-to-sequence design. RNA-FM adopts both the self-supervised learning approach and model architecture similar to BERT (Devlin et al., 2019), where around 15% of nucleotide tokens are randomly masked during training. The model is trained on 23 million sequences from the RNAcentral (rna, 2019) database, a dataset vastly larger than the available structure-sequence pairs. A masked language modeling (MLM) objective is employed, which minimizes the negative log-likelihood of the true nucleotide given the masked sequence context. This objective not only enables the model to predict the masked tokens accurately, but also allows it to learn intricate evolutionary information within its sequential input, as demonstrated by experimental validation on downstream tasks such as structure prediction.

## 3.2 STRUCTURE MODULE

We utilize the pre-trained RDesign (Tan et al., 2023) model for RNA tertiary structure modeling and initial coarse sequence prediction.

**RNA tertiary structure featurizer.** To model the complex three-dimensional folding of RNA molecules effectively, a comprehensive approach is essential. This approach involves representing RNA tertiary structure as an attributed graph $\mathcal{G} = (V, E)$, where $V$ denotes node attributes and $E$ denotes edge attributes. The graph is constructed by identifying the $K$ nearest neighbors in three-dimensional space for each nucleotide, with each nucleotide $i$ having a set of $K$ neighbors, denoted $\mathcal{N}(i, K)$. **Node attributes** $V \in \mathbb{R}^{N \times f_n}$ encompass $f_n$-dimensional features for N nucleotides, describing the local geometry of each nucleotide. These features involve dihedral angles, spatial distances, and directional vectors. **Edge attributes** $E \in \mathbb{R}^{N \times K \times f_m}$ include $f_m$-dimensional features for each nucleotide's $K$ neighbors, describing the relative geometry between nucleotides. These features include relative orientation, spatial distances, and directional vectors. Details on how to encode these attributes are provided in Sec.A.6.1

**MPNN layers.** Once the RNA tertiary structure is accurately modeled, $L$ layers of message-passing neural networks (MPNNs) are employed to learn the node representations. Specifically, the hidden state of the $i$-th nucleotide at the $l$-th layer is defined as follows:

$$\boldsymbol{h}_{V_i}^{(l)} = \text{MPNN}\left(\left[\boldsymbol{h}_{E_{i,j}}, \boldsymbol{h}_{V_i}^{(l-1)}, \sum_{j \in \mathcal{N}(i,K)} \boldsymbol{h}_{V_j}^{(l-1)}\right]\right) \qquad (2)$$

where $\boldsymbol{h}_V^{(0)}$, $\boldsymbol{h}_E$ are the embeddings of the node attributes and edge attributes from the tertiary structure modeling, respectively. Finally, a linear layer followed by a softmax function maps the output node representation $\boldsymbol{h}_{V_i}^L$ to the RNA sequence probabilities $\hat{p}(\mathcal{S}^N)$, which are then converted to the predicted RNA sequence $\hat{\mathcal{S}}^N$ using an argmax function.

$$\hat{p}(s_i) = \text{softmax}\left(\text{Linear}\left(\boldsymbol{h}_{V_i}^L\right)\right)$$
$$\hat{s}_i = \arg\max\left(\hat{p}\left(s_i\right)\right)$$

(3)

## 3.3 SEQUENCE REFINEMENT

As illustrated in Fig.2, we integrate evolutionary information from the pre-trained RNA language model, RNA-FM, to further refine RNA sequence prediction derived from the aforementioned structure module. This framework consists of three main components: weighted token embedding, structural adapter, and attention bias guidance.

**Weighted token embedding.** For each nucleotide $i$, RDesign predicts both the probabilities for 4 nucleotide categories, denoted as $\hat{p}(s_i) = \{p_i^j \mid 0 < p_i^j < 1\}_{j=0,\ldots,3}$, and the nucleotide type $\hat{s}_i$ itself. Correspondingly, RNA-FM provides word embeddings for these categories, represented as $\text{W}_{\text{FM}} = \{\boldsymbol{w}^j \in \mathbb{R}^{d_{\text{FM}}}\}_{j=0,\ldots,3}$. A trivial approach to generate the input token $\boldsymbol{h}_i^{\text{FM}}$ for RNA-FM is to directly use the word embedding corresponding to the predicted nucleotide type $\hat{s}_i$ from RDesign. Alternatively, we use the predicted probabilities $\hat{p}$ from RDesign to perform a weighted sum of the corresponding word embeddings $\text{W}_{\text{FM}}$ for producing the input token $\boldsymbol{h}_i^{\text{FM}}$ to RNA-FM, denoted as:

$$\boldsymbol{h}_i^{\text{FM}} = \sum_{j=0}^3 p_i^j \boldsymbol{w}^j$$

(4)

**Structural adapter.** Rather than directly feeding the token $\boldsymbol{h}_i^{\text{FM}}$ into RNA-FM, we adopt a basic transformer layer as the structural adapter. This ensures that structural features are infused while maintaining the model's simplicity. Specifically, the node embeddings $\boldsymbol{h}_V^{(0)}$ from the tertiary structure modeling are incorporated. This structural adapter composes three main components: (1) self-attention mechanism, which captures dependencies within the sequential data; (2) cross-attention mechanism, which retrieves relevant structural information from the structure encoder, bridging the gap between sequential and structural domain; (3) bottleneck feed-forward network (FFN), which introduces non-linearity and facilitates representation power.

$$\boldsymbol{h}_{\text{in}}^{\text{FM}} = \text{FFN}\left(\text{CrossAttn}\left(\text{SelfAttn}\left(\boldsymbol{h}^{\text{FM}}\right), \boldsymbol{h}_V^{(0)}\right)\right)$$

(5)

**Attention bias guidance.** To make the general-purpose RNA language model better retain the priors from the specialized structure-based RNA design model, and to ensure that the 3D structural interactions essential for structure-based RNA design are not overlooked during sequence refinement, we propose an attention bias guidance mechanism. Specifically, the edge features $\boldsymbol{h}_{E_{ij}}^{(L-1)}$ from RDesign are passed through a linear layer and added to the attention bias of all self-attention layers in the RNA-FM. Let $\boldsymbol{A}_{i,j}$ be the original attention scores. The updated attention scores $\boldsymbol{A}_{i,j}^{\text{update}}$ are:

$$\boldsymbol{A}_{i,j}^{\text{update}} = \boldsymbol{A}_{i,j} + \boldsymbol{A}_{i,j}^{\text{bias}} = \boldsymbol{A}_{i,j} + \text{Linear}\left(\boldsymbol{h}_{E_{ij}}^{(L-1)}\right)$$

(6)

## 4 EXPERIMENTS

In this section, we investigate three key questions through a series of experiments. (1) Whether language model can benefit structure-based RNA design (Sec.4.3). (2) Whether the proposed enhancement methods in Sec.3.3 are effective (Sec.4.4). (3) Whether the designed sequence can fold into a given 3D RNA structure (Sec.4.5). Before addressing these questions, we first introduce the benchmark (Sec.4.1) and baselines (Sec.4.2).

## 4.1 BENCHMARK

**Datasets.** In our research, we use the RNAsolo benchmark, proposed by RDesign (Tan et al., 2023), for training and evaluating. This dataset includes 2218 tertiary RNA structures in total, sourced

| Method | Short | Medium | Long | All |
|---|---|---|---|---|
| StructMLP | $25.72 \pm 0.51$ | $25.03 \pm 1.39$ | $25.38 \pm 1.89$ | $25.35 \pm 0.25$ |
| StructGNN | $27.55 \pm 0.94$ | $28.78 \pm 0.87$ | $28.23 \pm 1.95$ | $28.23 \pm 0.71$ |
| GraphTrans | $26.15 \pm 0.93$ | $23.78 \pm 1.11$ | $23.80 \pm 1.69$ | $24.73 \pm 0.93$ |
| PiFold | $24.81 \pm 2.01$ | $25.90 \pm 1.56$ | $23.55 \pm 4.13$ | $24.48 \pm 1.13$ |
| RDesign | $37.22 \pm 1.14$ | $44.89 \pm 1.67$ | $43.06 \pm 0.08$ | $41.53 \pm 0.38$ |
| EVO-RDesign | $\mathbf{43.68 \pm 1.08}$ | $\mathbf{46.40 \pm 0.31}$ | $\mathbf{44.93 \pm 0.46}$ | $\mathbf{44.93 \pm 0.43}$ |

Table 1: The sequence recovery(%) for RNAs of short, medium, and long sequences in the benchmark RNAsolo dataset.

from RNAsolo (Adamczyk et al., 2022) and the Protein Data Bank (PDB) (Berman et al., 2002). To address variations in RNA sequence lengths that may affect prediction outcomes, we follow RDesign to stratify the testing set into three length categories: (i) Short, for RNA samples of 50 nucleotides or fewer; (ii) Medium, for RNA samples between 51 and 100 nucleotides; (iii) Long, for RNA samples over 100 nucleotides. To further explore the zero-shot generation capabilities of EVO-RDesign, we utilize the RNA-Puzzles dataset (Miao et al., 2020), following previous work (Tan et al., 2023). This dataset serves as a common evaluation benchmark and has no overlap with training set. Further details are provided in the Sec.A.2.

**Evaluation metrics.** Following RDesign (Tan et al., 2023), we adopt sequence recovery and Macro-F1 scores to evaluate our model, suitable for RNA sequences that consist of only four types of nucleotides. The recovery score quantifies the accuracy of regenerating target RNA sequences, while the Macro-F1 score assesses performance by viewing RNA sequence design as a multi-class classification task. Detailed explanations of these metrics are available in Sec.A.1. We performed each experiment three times using different random seeds, reporting both mean and standard deviations.

## 4.2 BASELINES

**Tertiary structure-based methods.** We compare EVO-RDesign with various tertiary structure-based RNA sequence design models. StructMLP (Tan et al., 2023) utilizes structural features but overlooks graph topology. Conversely, StructGNN (Ingraham et al., 2019b), GraphTrans (Ingraham et al., 2019b), and PiFold (Gao et al., 2022) integrate graph topology. RDesign (Tan et al., 2023), a recent state-of-the-art method, additionally employs a hierarchical, data-efficient contrastive learning framework.

**Secondary structure-based methods.** We also benchmark methods that take the secondary structures of RNA as input. MCTS-RNA (Yang et al., 2017) employs the method of Monte Carlo tree search to explore viable designs, while LEARNA (Runge et al., 2018) utilizes a deep reinforcement learning approach. Additionally, aRNAque (Merleau & Smerlak, 2022) and eM2dRNAs (Rubio-Largo et al., 2023) introduce evolutionary algorithms specifically tailored for secondary structure-based RNA design.

## 4.3 MAIN RESULTS

**Standard evaluation.** We utilize the RNAsolo benchmark for training and evaluation, adhering to the train/validation/test dataset split of RDesign (Tan et al., 2023). We adjust our model design and hyper-parameters based on validation set performance and report the evaluation results on the test set. EVO-RDesign achieves significantly better performance across all metrics (recovery and Macro-F1) and all length ranges, as presented in Tab.1 and Tab.2. Compared to our structure module counterpart RDesign, EVO-RDesign exhibits a 3.5% enhancement in the sequence recovery metric, underscoring the substantial benefit of incorporating the RNA language model in structure-based RNA design. It is noteworthy that EVO-RDesign achieves greater performance gains on shorter sequences, which is attributed to the pre-trained RNA language model. RNA-FM (Chen et al., 2022) shows a decline in performance as sequence length increases in both secondary and tertiary structure prediction tasks.

| Method | Short | Medium | Long | All |
|---|---|---|---|---|
| StructMLP | $17.46 \pm 2.39$ | $18.57 \pm 3.45$ | $17.53 \pm 8.43$ | $18.88 \pm 2.50$ |
| StructGNN | $24.01 \pm 3.62$ | $22.15 \pm 4.67$ | $26.05 \pm 6.43$ | $24.87 \pm 1.65$ |
| GraphTrans | $16.34 \pm 2.67$ | $16.39 \pm 4.74$ | $18.67 \pm 7.16$ | $17.18 \pm 3.81$ |
| PiFold | $17.48 \pm 2.24$ | $18.10 \pm 6.76$ | $14.06 \pm 3.53$ | $17.45 \pm 1.33$ |
| RDesign | $38.25 \pm 3.06$ | $40.41 \pm 1.27$ | $41.48 \pm 0.91$ | $40.89 \pm 0.49$ |
| EVO-RDesign | $\mathbf{41.39 \pm 0.39}$ | $\mathbf{46.21 \pm 0.18}$ | $\mathbf{43.72 \pm 0.30}$ | $\mathbf{44.37 \pm 0.18}$ |

Table 2: The Macro-F1($\times 100$) for RNAs of short, medium, and long sequences in the benchmark RNAsolo dataset.

Therefore, a plausible explanation for our experimental observations is that RNA-FM, the pre-trained RNA language model we choose, possesses stronger priors for shorter sequences.

**Zero-shot generalization.** To further explore the zero-shot generalization capabilities of EVO-RDesign, we conduct evaluations on the Rfam (Kalvari et al., 2021) and RNA-Puzzles dataset (Miao et al., 2020). It is worth noting that the results presented here are obtained by directly evaluating pre-trained models on unseen structure-based RNA design tasks, reflecting real-world conditions in wet-lab scenarios. We present the results in Tab.3, in which the performance remains consistent with that of RNAsolo benchmark dataset. Specifically, our proposed EVO-RDesign model demonstrates superior generalization across all metrics, and outperforms all the baselines by a large margin, whether they are secondary-based methods or tertiary-based methods. Compared to our structure module counterpart RDesign in the sequence recovery metric, EVO-RDesign achieves 5.1% and 4.1% increases in sequence recovery on RNA-Puzzles and Rfam, respectively, demonstrating its zero-shot generalization capabilities.

| Method | Input Structure | Recovery (%) ↑ | | Macro F1 ($\times 100$) ↑ | |
|---|---|---|---|---|---|
| | | Rfam | RNA-Puzzles | Rfam | RNA-Puzzles |
| MCTS-RNA | Secondary | $31.74 \pm 0.07$ | $32.06 \pm 1.87$ | $23.82 \pm 4.60$ | $24.12 \pm 3.47$ |
| LEARNA | Secondary | $31.92 \pm 2.37$ | $30.94 \pm 4.15$ | $24.02 \pm 3.73$ | $22.75 \pm 1.17$ |
| aRNAque | Secondary | $30.01 \pm 3.26$ | $31.07 \pm 2.32$ | $22.84 \pm 1.70$ | $23.30 \pm 1.65$ |
| eM2dRNAs | Secondary | $33.34 \pm 1.02$ | $37.10 \pm 3.24$ | $24.80 \pm 3.88$ | $26.91 \pm 2.32$ |
| StructMLP | Tertiary | $24.40 \pm 1.63$ | $24.22 \pm 1.28$ | $16.79 \pm 4.01$ | $16.40 \pm 3.28$ |
| StructGNN | Tertiary | $27.64 \pm 3.31$ | $27.96 \pm 3.08$ | $24.35 \pm 3.45$ | $22.76 \pm 3.19$ |
| GraphTrans | Tertiary | $23.81 \pm 2.57$ | $22.21 \pm 2.98$ | $17.32 \pm 5.28$ | $17.04 \pm 5.36$ |
| PiFold | Tertiary | $22.55 \pm 4.13$ | $23.78 \pm 6.52$ | $16.08 \pm 2.34$ | $16.20 \pm 3.49$ |
| RDesign | Tertiary | $56.12 \pm 1.03$ | $50.12 \pm 1.07$ | $53.27 \pm 1.28$ | $49.24 \pm 1.07$ |
| EVO-RDesign | Tertiary | $\mathbf{60.22 \pm 0.27}$ | $\mathbf{55.17 \pm 0.18}$ | $\mathbf{58.25 \pm 0.31}$ | $\mathbf{55.16 \pm 0.11}$ |

Table 3: Zero-shot generation results on RNA-Puzzles and RFam. Native sequence recovery (%) and Macro F1 ($\times 100$) results of models.

### 4.4 ABLATION STUDY

**Design choices.** To verify the effectiveness of our proposed enhancement methods in Sec.3.3, we conducted ablation experiments, as shown in Tab.4. (1) **Without language model**: we eliminate the sequence refinement module that employs the RNA language model and directly output the RDesign prediction. (2) **Without weighted token embedding**: instead of using the weighted token embedding, we directly feed the word embedding corresponding to the RDesign predicted nucleotide type into the sequence refinement module. (3) **Without structural feature**: we remove the injection of tertiary structural features in the structural adapter. (4) **Without attention bias**: we remove the attention bias guidance mechanism, which guides the RNA language model to retain the priors from the specialized structure-based RNA design model. Removing any component (language model, weighted token embedding, structural features, or attention bias) results in lower performance, with the largest drop

| Method | Recovery (%) ↑ | | Macro F1 (×100) ↑ | |
|---|---|---|---|---|
| | RNAsolo | RNA-Puzzles | RNAsolo | RNA-Puzzles |
| w/o language model | $41.53 \pm 0.38$ | $50.12 \pm 0.18$ | $40.89 \pm 0.49$ | $49.24 \pm 1.07$ |
| w/o weighted token embedding | $43.92 \pm 0.56$ | $53.31 \pm 0.29$ | $44.03 \pm 0.32$ | $52.96 \pm 0.31$ |
| w/o structural feature | $44.34 \pm 0.52$ | $54.17 \pm 0.21$ | $44.25 \pm 0.10$ | $54.22 \pm 0.16$ |
| w/o attention bias | $44.82 \pm 0.14$ | $54.66 \pm 0.12$ | $44.19 \pm 0.19$ | $54.43 \pm 0.13$ |
| EVO-RDesign | $44.93 \pm 0.43$ | $55.17 \pm 0.18$ | $44.37 \pm 0.18$ | $55.16 \pm 0.11$ |

Table 4: Ablation of design choices. Native sequence recovery (%) and Macro F1 (×100) results of different design choices.

observed when the language model is excluded. These results clearly demonstrate the effectiveness of each proposed component.

**Training choices.** We evaluate the impact of various fine-tuning and freezing strategies on the model's performance in terms of Recovery and Macro F1 scores, as shown in Tab.5. The results indicate that **freezing both the RDesign and RNA-FM modules yields the best performance**, achieving the highest Recovery and Macro F1 scores. In contrast, fine-tuning either or both modules tends to result in lower performance, even when using LoRA (Hu et al., 2021) to fine-tune RNA-FM. LoRA (Low-Rank Adaptation) is a technique designed to preserve the pre-trained model's priors while reducing the computational complexity of fine-tuning large models by introducing low-rank updates, yet it still does not surpass the performance of the frozen configuration in this case. Our observations suggest that for RNA, where high-quality structure-sequence pair data is much scarcer than sequence data, the best way to leverage the priors of sequence models is to freeze them, thereby minimizing any disruption to the pre-trained models' priors. This approach ensures that the valuable evolutionary information captured from RNA sequences is preserved, leading to better performance compared to fine-tuning strategies that may introduce noise or overfit the limited available data. Details of training hyper-parameters are in Sec.A.3.

| RDesign | RNA-FM | Recovery (%) ↑ | Macro F1 (×100) ↑ |
|---|---|---|---|
| ✓ | ✗ | $41.44 \pm 2.46$ | $41.54 \pm 1.55$ |
| ✗ | LoRA | $43.36 \pm 0.46$ | $42.13 \pm 0.88$ |
| ✓ | LoRA | $42.49 \pm 3.25$ | $41.17 \pm 3.96$ |
| ✗ | ✗ | $44.93 \pm 0.43$ | $44.37 \pm 0.18$ |

Table 5: Ablation of training choices on the RNAsolo dataset. ✓, ✗, and LoRA indicate that the model is fine-tuned, frozen, and fine-tuned with LoRA, respectively.

## 4.5 IN-SILICO VALIDATION

We conduct our in-silico folding validation on the RNAsolo test set. Using RhoFold (Shen et al., 2022), we predict the tertiary structures of RNA sequences generated by EVO-RDesign, verifying their ability to fold into a given RNA 3D backbone. To assess the similarity between the predicted RNA structures and the given fixed backbones, we use the TM-score (Zhang & Skolnick, 2004) of C3' and C4' backbone atoms. As depicted in Fig.3, the folding results of sequences produced by EVO-RDesign are promising when compared to those generated by RDesign. In addition, the performance for long RNA sequences is relatively poor for both RDesign and EVO-RDesign. There are two possible reasons for this: the imbalance of different length data in our training set and the flexibility of RNA structure and conformation. RNA's structural intricacy and flexibility (Hagerman, 1997; Bernstein et al.; Townshend et al., 2021), which exceed those of proteins, make structure alignment difficult. As the length of the structures increases, it becomes progressively more difficult to achieve an accurate alignment between two highly flexible entities using a rigid-body transformation. This is because such transformations inherently lack the flexibility needed to account for the structural variations that naturally arise in longer sequences. This challenge is particularly relevant to the

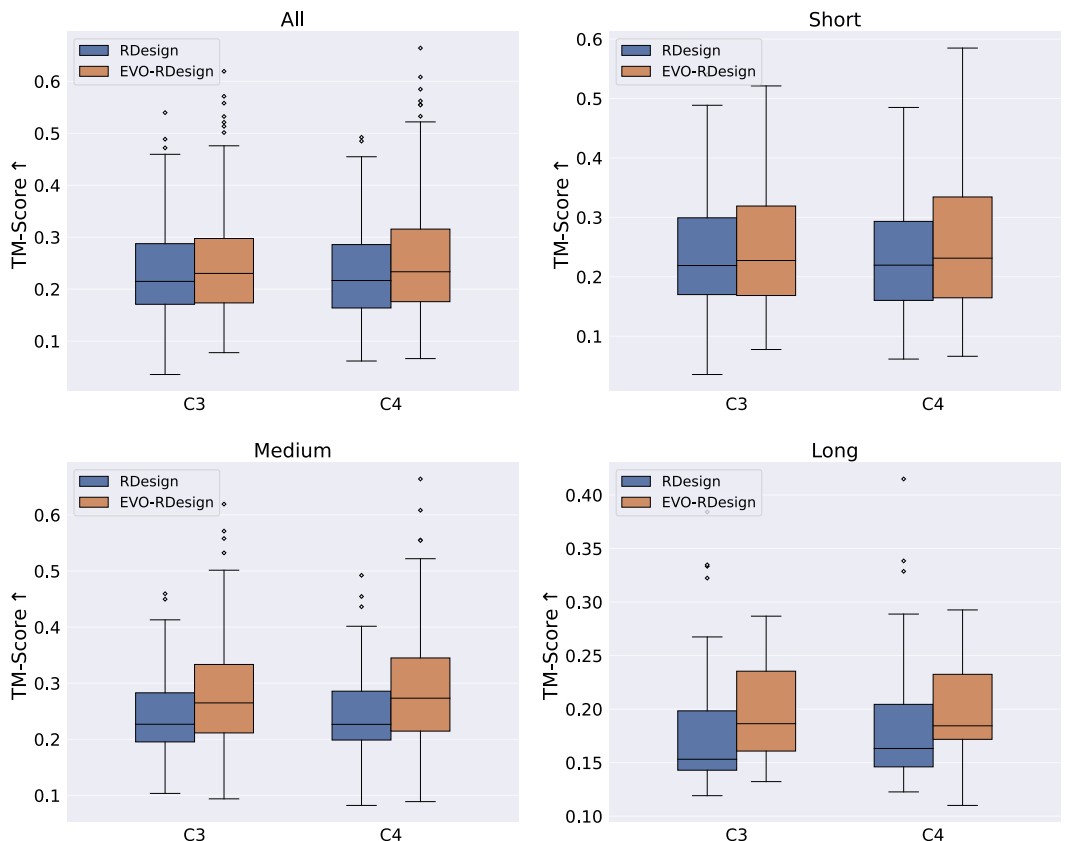

Figure 3: Visualization of in-silico tertiary structure folding validation. Each figure shows the boxplot of TM-score for all, short, medium, and long RNA sequences, respectively.

TM-score metric, which calculates the structural similarity between two protein models by using rigid transformation, making it less effective for longer, more flexible structures.

## 5 DISCUSSION AND CONCLUSION

We propose EVO-RDesign, a model-agnostic and generic approach that repurposes RNA language model for structure-based RNA design. Extensive experiments demonstrate the effectiveness of our proposed EVO-RDesign model. Through comprehensive analyses, we verify that EVO-RDesign (1) effectively exploits both structural and sequential knowledge and (2) generalizes to unseen structure-to-sequence RNA design tasks. However, given the early development stages of structure encoding and sequence modeling in the RNA field, our options are currently limited. This limitation prevents us from systematically experimenting with various structure encoders and language models, highlighting the need for collective efforts from the entire field in the future. RNA design has significant value for diagnostics, therapeutics, and synthetic biology. The ultimate goal of RNA design is to create functional RNAs, such as ribozymes, riboswitches, and aptamers, that can be validated through wet-lab experiments. To fully realize the potential of RNA design, it is essential to explore more advanced language model techniques, such as multimodal approaches for structure-sequence co-design. We hope our work provides a new perspective on tertiary structure-based RNA design and encourages further advancements in this field.

The **limitation** is that we have only conducted in-silico validation, rather than wet-lab validation. The **positive societal impact** is that this work can accelerate the RNA design process potentially. No **negative societal impact** is perceived.

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

## A APPENDIX

### A.1 METRIC

#### A.1.1 RECOVERY

We define the recovery metric for assessing the nucleotide-level prediction accuracy of an RNA sequence model as:

$$\text{Recovery}\left(\mathcal{S}^N, \mathcal{X}^N\right) = \frac{1}{N} \sum_{i=1}^{N} \mathbb{1}\left[\mathcal{S}_i^N = \arg\max p\left(\mathcal{S}_i^N \mid \mathcal{X}\right)\right]$$

This formula calculates the proportion of nucleotides that are correctly predicted as the most probable by the model, averaged across all $N$ positions. Here, $p\left(\mathcal{S}_i^N \mid \mathcal{X}\right)$ is the probability of correctly predicting nucleotide $\mathcal{S}_i^N$ at position $i$ based on the input structure $\mathcal{X}$, and $\mathbb{1}[\cdot]$ is the indicator function. This metric effectively measures how well the model predicts the RNA sequence at each nucleotide position.

### A.1.2 MACRO F1 SCORE

The Macro-F1 score is defined as the average of the F1 scores for each class in a multi-class classification problem.

$$\text{Macro-F1} = \frac{1}{|C|} \sum_{c \in \{\text{A,U,C,G}\}}^{C} 2 \times \frac{\text{Precision}_c \times \text{Recall}_c}{\text{Precision}_c + \text{Recall}_c}$$

Here, $|C|$ represents the number of classes, and the F1 score for each class c is the harmonic mean of precision and recall for each class. The Macro-F1 score effectively evaluates the model's performance across all classes, taking into account both the precision and recall, which makes it particularly useful for datasets with class imbalances. It assesses the model's ability to correctly identify instances of each class.

## A.2 DATASET

### A.2.1 RNASOLO

We have adopted the RNAsolo benchmark, initially proposed by RDesign (Tan et al., 2023), for training and evaluating our models. This benchmark dataset combines and refines data from RNA-solo (Adamczyk et al., 2022) and the Protein Data Bank (PDB) (Berman et al., 2002), providing a comprehensive and robust framework for assessing RNA structure prediction algorithms. The dataset includes 2218 tertiary RNA structures, thoughtfully divided to maintain sequence length consistency and minimize distribution shifts across subsets. It consists of 1774 structures for training, 223 for testing, and 221 for validation, grouped by structural similarities. To prevent information leakage and ensure uniform evaluations, the dataset is organized into 987 clusters. These clusters are strategically allocated to specific sets, with larger clusters containing over 30 samples assigned to the training set to avoid biases in the smaller test and validation sets. This approach enhances the reliability of our model assessments.

### A.2.2 RNAPUZZLES

Introduced by Miao et al. (2020), the RNA-puzzles dataset is a publicly available series of challenges designed to test and enhance the precision of algorithms used in RNA structure prediction. Each challenge in this dataset includes experimental data, notably X-ray crystallography results formatted in PDB, coupled with a specific target sequence and the corresponding RNA structure. This dataset serves as a critical tool for benchmarking and evaluating the performance of various computational strategies for predicting RNA configurations. We specifically employ this dataset to assess the capabilities of EVO-RDesign in structure-based RNA design.

## A.3 TRAINING

### A.3.1 HYPER-PARAMETERS

We train the model over 10 epochs using the Adam optimizer, with a learning rate set at 0.001 and a batch size of 16. For the structure module, we follow RDesign's parameter choices. With a dropout rate of 0.1, it considers 30 nearest neighbors and a vocabulary size matching RNA's four alphabets. For the RNA language model, we use the pre-trained RNA-FM model exclusively, with an embedding dimension of 640.

For the ablation study involving LoRA fine-tuning of RNA-FM, we tested various combinations of configurations with lora rank set to [4, 8, 16, 32, 64, 128], alpha set to [4, 8, 16, 32, 64, 128], and dropout ratio set to [0, 0.25, 0.5, 0.75]. We report the results with the best set of parameters. All

experiments are repeated five times with different random seeds, and we report the mean and standard deviation.

### A.3.2 HARDWARE

All our experiments are conducted on a computing cluster with 8 GPUs of NVIDIA GeForce RTX 4090 24GB and CPUs of AMD EPYC 7763 64-Core of 3.52GHz. All the inferences are conducted on a single GPU of NVIDIA GeForceRTX 4090 24GB.

### A.4 ADDITIONAL RESULTS

**In-silico folding validation.** We perform our in-silico validation on the RNAsolo test set. Tab.6 describes the mean TM-Score for different RNA sequence length ranges, where EVO-RDesign surpasses RDesign in all cases.

Table 6: In-silico tertiary structure folding validation on RNAsolo dataset.

| Method | Mean TM-Score on C3'↑ | | | | Mean TM-Score on C4'↑ | | | |
|---|---|---|---|---|---|---|---|---|
| | All | Short | Medium | Long | All | Short | Medium | Long |
| RDesign | 0.2315 | 0.2335 | 0.2479 | 0.1883 | 0.2317 | 0.2315 | 0.2497 | 0.1918 |
| EVO-RDesign | 0.2506 | 0.2428 | 0.2887 | 0.1948 | 0.2620 | 0.2571 | 0.2984 | 0.1978 |

### A.5 LORA

As language models grow in size and complexity, the resource demands for fine-tuning these models become increasingly prohibitive. To address this, LoRA (Low-Rank Adaptation) introduces a method for efficient adaptation by incorporating trainable low-rank matrices into the pre-trained model's layers, while keeping the original model weights static (Hu et al., 2021).

LoRA modifies the weight matrix $\mathbf{W} \in \mathbb{R}^{d \times k}$ of the Transformer layers by adding a low-rank update. The adaptation is formulated as follows:

$$\mathbf{W} \leftarrow \mathbf{W} + \mathbf{A}\mathbf{B}, \tag{7}$$

where $\mathbf{A} \in \mathbb{R}^{d \times r}$ and $\mathbf{B} \in \mathbb{R}^{r \times k}$ are trainable matrices with $r$ being much smaller than both $d$ and $k$.

LoRA presents multiple advantages, including a drastic reduction in the number of parameters that need to be trained, with reductions reaching up to 10,000 times less than what is required for full model fine-tuning. It also substantially cuts down on GPU memory usage, requiring only a third of the memory compared to traditional methods. Performance-wise, LoRA meets or exceeds the results of full fine-tuning on various benchmarks such as RoBERTa, DeBERTa, GPT-2, and GPT-3. Additionally, unlike methods that rely on adapters, LoRA does not introduce any extra latency during inference. In summary, LoRA offers an efficient and effective approach to adapting large pre-trained language models to new tasks. By utilizing low-rank adaptations, it significantly alleviates the computational and memory demands of fine-tuning while maintaining or even enhancing model performance.

### A.6 RDESIGN

### A.6.1 ATTRIBUTES ENCODING

Node attributes $V \in \mathbb{R}^{N \times f_n}$ encompass $f_n$-dimensional features for N nucleotides, describing the local geometry of each nucleotide. These features involve: (1) dihedral angles, represented using sine and cosine functions; (2) spatial distances, encoded into radial basis functions (RBFs) relative to a reference atom $P_i$; (3) directional vectors, calculated with respect to the local coordinate system $Q_i$.

Edge attributes $E \in \mathbb{R}^{N \times K \times f_m}$ include $f_m$-dimensional features for each nucleotide's $K$ neighbors, describing the relative geometry between nucleotides. These features include: (1) orientation encoding, derived from the quaternion representation of the relative rotation between $Q_i$ and $Q_j$; (2)

spatial distances, encoded into RBFs between inter-nucleotide atoms and the reference atom $P_i$ ; (3) directional vectors, computed relative to the reference atom $P_i$.

