# OpenReview forum: "EVO-RDesign: Leveraging Evolutionary Priors for Structure-Based RNA Design"
_ICLR.cc/2025/Conference — Submitted to ICLR 2025_

### Official Review · Reviewer_SmLw · 2024-10-20

**Soundness:** 2
**Presentation:** 3
**Contribution:** 2
**Rating:** 5
**Confidence:** 5

**Summary:**

The paper proposed an RNA design approach that learns the evolutionary information of RNA, which outperforms the previous approaches. Namely, it proposed a sequence refinement module in addition to GNN layers. The paper is clearly written and easy to follow.

**Strengths:**

- The experiment is solid, considering various sequence lengths, which provides a more comprehensive insight for the reader.
- This paper enriches the less-explored field of RNA design, which might be crucial for improving biotechnologies like CRISPR
- It is an interesting approach to use evolutionary information.

**Weaknesses:**

Not the first time reviewing this paper, as my previous questions were not answered at all, some questions persisted. I would strongly recommend the authors go through previous reviews and make necessary modifications and clarifications before submitting to another conference.
- Why does the RNA structural featurizer have to go through a transformer layer, as stated in Equation 5, to be the input of the RNA language model? Since it is the most important part of this paper, I think it is worth more discussion.
- The paper uses structural features from the RDesign model and designed a structural adapter as the input of RNA language model, which means the key information is just retained from those two pre-trained models and, maybe less significant to the RNA design field.
- The paper uses RNA sequential evolutionary information before the sequence output, such design may have information leakage since the RNA language model is trained with raw sequences.

**Questions:**

- My main question would be the information leakage issue, since the RNA language model is trained with sequences, is it proper to put it in the output layer of RNA sequence design model? For example, if the RNA-FM model has "seen" a similar sequence/embedding during its training phase, its output may tend to be like that sequence. However, as the recovery score(or accuracy) is less than half (overall), would such probabilities output mislead the RNA-FM model? Even more, when the input structure is not contained in the training dataset of RNA-FM, would the model give a relatively worse output than others? I suggest the authors have a more detailed clarification on this issue.
- Another question is about the in-silico validation. The paper uses RhoFold to test whether the generated sequence could fold back into the original structure. It is indeed a pretty hard goal because RNA structure prediction tools are still unable to provide reliable predictions. - However, it might be helpful to have a quantitive analysis of the samples. For example, what kind of RNAs always have a better result than others, and what kind of sequence length may have the best results... Also, conducting such experiments on external datasets may also be helpful (as tested in the paper, RNA-puzzles)
- Some other external datasets could be tested, such as Rfam, which has been used in the RDesign paper.
Would other metrics also be helpful to evaluate the model's capability? Such as AUROC and AUPRC.

**Details Of Ethics Concerns:**

No related concerns

---

### Official Review · Reviewer_k9nG · 2024-10-28

**Soundness:** 2
**Presentation:** 3
**Contribution:** 1
**Rating:** 3
**Confidence:** 4

**Summary:**

The paper describes EVO-RDesign, a method for structure-based RNA design from 3D RNA backbone inputs. The authors use predictions from an existing 3D RNA design algorithm, RDesign, which are then fed into a pre-trained RNA foundation model, RNA-FM, using adapter modules to include structure information into the foundation model. In the results, the authors show improved performance compared to RDesign and several other methods based on sequence recovery.

**Strengths:**

- The combination of the foundation model with the previous work seems to result in a performance improvement compared to the original design algorithm.
- The datasets used seem to be appropriate regarding data processing based on sequence and structure similarity.
- The idea of using evolutionary information for structure-based RNA design is interesting.

**Weaknesses:**

- The contributions of this work are very minor, with the main contribution being the implementation of a module that combines two existing pre-trained models by conditioning the foundation model on the structure.
- The model is trained on sequence recovery which doesn’t make much sense for the use case of RNA design in general (see questions). However, when evaluating structure recovery as shown in Figure 3, the resulting average TM- (at roughly 0.2-0.3) seems to be rather poor.

**Questions:**

- Why did the authors not release the code? I cannot see any reason for this since most of the work is based on two existing models and the adapter modules mainly link the two?
- Computational structure-based RNA Design is generally used to find promising candidates for subsequent analysis in the wet-lab. The provided candidates thus should be diverse while folding into the desired structure (a 3D backbone in this case). While I know that this has been done before, why do we train on sequence recovery then at all? This means that we are good at replicating the sequences we already know to fold into the structure. At least a measure of diversity would make sense to assess the distribution of generated sequences.
- While the authors describe gRNAde as a related work, they do not include it into the evaluation? Why?

---

### Official Review · Reviewer_Wb7J · 2024-11-02

**Soundness:** 3
**Presentation:** 2
**Contribution:** 2
**Rating:** 3
**Confidence:** 4

**Summary:**

This paper introduces EVO-RDesign, a framework for structure-based RNA design that leverages RNA language models. The paper proposes a method to map structural information into a format acceptable by the language model, enabling it to predict RNA sequences directly based on input structural information. The EVO-RDesign framework consists of three main components: weighted token embedding, structural adapter, and attention bias guidance. The paper evaluates the performance of EVO-RDesign on two benchmark datasets, RNAsolo and RNA-Puzzles, demonstrating its superior generalization across all metrics compared to baseline methods.

**Strengths:**

1 .The paper evaluates the performance of EVO-RDesign on two benchmark datasets, providing a comprehensive comparison with baseline methods.
2. EVO-RDesign demonstrates superior generalization across all metrics compared to baseline methods, highlighting its effectiveness in structure-based RNA design。

**Weaknesses:**

1. While the paper provides an overview of the EVO-RDesign framework, some technical details, such as the specific implementation of the structural adapter and attention bias guidance, are not fully explained. It is hard to follow. Provide more detailed explanations of the technical components of EVO-RDesign, such as the structural adapter and attention bias guidance, to facilitate better understanding and reproducibility.
2. How does EVO-RDesign handle large-scale RNA structures? Are there any limitations in terms of the size or complexity of the RNA structures that can be designed using EVO-RDesign? What is the mean size of RNA sequences that you used or produced?
3. The authors stated that "EVO-RDesign,which leverages the evolutionary priors", what can of prior information that you used? Could you explain these priors more? or give examples in details?
4. In the abstract, the authors say they also apply in-silico folding, I guess it is better to add the conclusions of applying in-silico folding.
5. The evolutionary information contained in RNA sequences, but this method used RNA structures as input, I deem the evolutionary information is already contained in the structures. More, in Fig.2, the important concept "evolutionary priors" is not appeared.
6. In Fig, RNA consists of four types of nucleotide, what are the four types of nucleotide?
7. In Line 76-77, the authors say they use RNA language model first, and then use the structure model, which is  consistent with Fig.2. More, what are FFN and MPNN in Fig.2.
8.In Sec.1, the authors say "However, these methods still face significant bottlenecks: the known RNA structure data is very limited." However, this method also faces this problem, as the model needs RNA structures as input.
9. In Line 202-203, what is the K values in the K-nearest neighbors? What is the choose standard of K?
10.Why you choose RDesign to extract the structural features, have you tried other structure encoders? Similar problem for the RNA Language Model.
11. What is your future plan wet-lab validation?

**Questions:**

See Weakness.

---

### Official Review · Reviewer_RcFS · 2024-11-03

**Soundness:** 2
**Presentation:** 2
**Contribution:** 1
**Rating:** 3
**Confidence:** 4

**Summary:**

I will cut directly into strengths/weakness/questions.

**Strengths:**

This paper introduced a new upgrade to RDesign in that now evolutionary information has been added to the model.

**Weaknesses:**

- This is yet another RDesign but with an adapter introduced to bring in evolutionary information from a pretrained RNA LLM. The metholdological novelty is limited.
- Despite what has already been shown in gRNAde, the dataset used in this paper for training and evaluation still run the risk of information leak/contamination. This is very important because this essentially puts a question mark on all the results.
- The evaluation metrics reported for all but the last subsections in the result section have only looked at sequence recovery rate which is unable to capture the quality on the structural level.
- Also, given the performance issue that has been discussed in gRNAde, this paper still only compares to RDesign as the main deep learning competitor.
- The result section does not offer any valuable insights biologically, except for a benchmark which is quite boring.

**Questions:**

- How does the inclusion of LLM aid the inverse design of RNA aptamers? This is some biology worth digging deeper into.

---

### Meta-Review · Area_Chair_zvMe · 2024-12-31

**Metareview:**

The idea of using evolutionary information in structure-based RNA design is attractive. However, the reviewers have major concerns about the experimental design, the quality of writing, and the overall contribution (see the reviews for more details), and there is no author response. Given this, I am recommending rejection.

**Additional Comments On Reviewer Discussion:**

The authors have not submitted rebuttal comments.

---

### Decision · Program_Chairs · 2025-01-22

Reject